# Treatment-Resistant Depression (TRD): Is the Opioid System Involved?

**DOI:** 10.3390/ijms241311142

**Published:** 2023-07-06

**Authors:** Shaul Schreiber, Lee Keidan, Chaim G. Pick

**Affiliations:** 1Department of Psychiatry, Tel Aviv Sourasky Medical Center, Tel Aviv 6423906, Israel; shaulsch@tauex.tau.ac.il; 2Dr. Miriam and Sheldon G. Adelson Clinic for Drug Abuse Treatment and Research, Tel Aviv Sourasky Medical Center, Tel Aviv 6423906, Israel; 3Faculty of Medicine and Sagol School of Neuroscience, Tel Aviv University, Tel Aviv 6997801, Israel; 4Sylvan Adams Sports Institute, Tel Aviv University, Tel Aviv 6905904, Israel; leekeidan@mail.tau.ac.il; 5Department of Anatomy and Anthropology, Tel Aviv University, Tel Aviv 6905904, Israel; 6Dr. Miriam and Sheldon G. Adelson Chair and Center for the Biology of Addictive Diseases, Tel Aviv University, Tel Aviv 6905904, Israel

**Keywords:** treatment-resistant depression, opioids, mice, antinociception, mianserin, mirtazapine, trazodone, venlafaxine, reboxetine, moclobemide, fluoxetine, fluvoxamine

## Abstract

About 30% of major depression disorder patients fail to achieve remission, hence being diagnosed with treatment-resistant major depression (TRD). Opium had been largely used effectively to treat depression for centuries, but when other medications were introduced, its use was discounted due to addiction and other hazards. In a series of previous studies, we evaluated the antinociceptive effects of eight antidepressant medications and their interaction with the opioid system. Mice were tested with a hotplate or tail-flick after being injected with different doses of mianserin, mirtazapine, trazodone, venlafaxine, reboxetine, moclobemide, fluoxetine, or fluvoxamine to determine the effect of each drug in eliciting antinociception. When naloxone inhibited the antinociceptive effect, we further examined the effect of the specific opioid antagonists of each antidepressant drug. Mianserin and mirtazapine (separately) induced dose-dependent antinociception, each one yielding a biphasic dose-response curve, and they were antagonized by naloxone. Trazodone and venlafaxine (separately) induced a dose-dependent antinociceptive effect, antagonized by naloxone. Reboxetine induced a weak antinociceptive effect with no significant opioid involvement, while moclobemide, fluoxetine, and fluvoxamine had no opioid-involved antinociceptive effects. Controlled clinical studies are needed to establish the efficacy of the augmentation of opiate antidepressants in persons with treatment-resistant depression and the optimal dosage of drugs prescribed.

## 1. Introduction

Major depressive disorder (MDD) is a severe mental disorder with a lifetime risk of experiencing an episode estimation of close to 1 in 5 people. At any given time, it affects approximately 350 million people worldwide and is one of the leading causes of mental disability [1]. Conventional antidepressant treatment is based on the monoaminergic hypothesis, and thus the vast majority of antidepressants target monoaminergic (serotonin, noradrenaline, and dopamine) receptors [2]. It has been estimated that about 30% of MDD patients do not benefit from pharmacotherapy even when given multiple antidepressants and applying various augmentation strategies [1]. This substantial portion of patients who fail to achieve remission despite the use of at least two adequately dosed antidepressant regimens [for a long enough time] are termed as having treatment-resistant major depression (TRD) [3,4]. TRD patients have significantly more outpatient visits and are at least twice as likely to be hospitalized [5]. TRD is, therefore, a costly disease associated with the extensive use of depression-related and general medical services. 

For centuries, opium had been largely used to treat depression and was considered an effective treatment not only for melancholia (an old term for depression) but for refractory melancholia (nowadays referred to as ‘treatment resistant depression’, TRD) as well [6,7,8]. However, opium is highly addictive, and many patients develop dependence and addiction. Moreover, following extensive diversion, it has become a widespread street drug (in a way, resembling the current opioid crisis in the USA and some other Western countries, including Israel) [9,10,11]. 

To date, several types of opioid receptors have been discovered: the m (mu) receptor (MOR), k (kappa) receptor (KOR), d (delta) receptor (DOR), and nociceptin/orphanin FQ receptor (NOR), while other types have been proposed (e.g., s (sigma), e (epsilon), and z (zeta) opioid receptors are being studied). Within these different types are a subset of subtypes (i.e., m1, m2, k1, etc.) [12].

Since the 1950s, following the serendipitous discovery of the antidepressant properties of mono-amine oxidase inhibitors (MAO-Is) and the development of tricyclic antidepressants (TCAs), the use of opioids in the treatment of depression has been discontinued. The general notion was that those ‘new’ classes of medications possess no opioid properties and are not addictive. But a growing body of evidence indicates the direct involvement of the endogenous opioid system in the brain pathophysiology of depression [13,14], and the possible association of the opioid system in the mechanism of action of some antidepressants has returned to focus. 

Studies using rodents that look at the possible interactions between some tricyclic antidepressants (amoxapine, amitriptyline nortriptyline, desipramine, and imipramine) and the opioid receptors of some tissue cells found, indeed, that these drugs interact with the delta opioid receptors (amoxapine exerting the highest potency and efficacy) and the kappa opioid receptors (amitriptyline exerting the strongest effect). However, all tested drugs showed low affinity and no antagonist activity at the mu-opioid receptors. The researchers conclude that “These results show that TCAs differentially regulate opioid receptors with a preferential agonist activity on either delta or kappa subtypes and suggest that this property may contribute to their therapeutic and/or side effects” [15]. In another study, naloxone (a preferential mu-opioid receptor antagonist) suppressed the behavioral activity of imipramine but not that of desipramine or amineptine [16]. 

Due to the significant side effects of tricyclic antidepressant medications, a myriad of compounds with antidepressant efficacy have been developed and launched over the decades, aiming to achieve better tolerability to their unwanted effects. 

Mianserin (launched in Europe in 1979) is a nonselective tetracyclic antidepressant with a prominent serotonergic and moderate noradrenergic profile and with moderate histaminergic side effects. It strongly blocks post-synaptic 5-HT receptors and only weakly blocks post-synaptic 5-HT and 5-HT1,3 receptors [17,18,19] and blocks moderately presynaptic a2 receptors. Mianserin’s main side effects derive from its antihistamine (H1) effects. Mianserin and its active metabolites have a half-life of 24–32 h in plasma; it lacks anticholinergic side effects, and it is much less cardiotoxic than tricyclic [20,21].

Mirtazapine is one of a series of chemical compounds known as piperazinoazepines and is not related to any known class of psychotropic drugs. Mirtazapine enhances noradrenergic and 5-HT_1A_-mediated serotonergic neurotransmission via the antagonism of central α_2_-auto- and hetero-adrenoreceptors [22,23]. It does not inhibit noradrenaline or serotonin uptake, but it specifically blocks the 5-HT_2_- and 5-HT_3_-type receptors while failing to modulate monoamine reuptake in animal models. In these models, mirtazapine manifested a very low affinity for dopaminergic receptors and a high affinity for histamine H_1_ receptors [23]. Mirtazapine’s unique pharmacological profile, which, in part, resembles that of mianserin [24], suggests a combined serotonin-noradrenaline mediated antinociception with a possible involvement of the opioid system. 

Trazodone is a triazolopyridine derivative with antidepressant activity and has been in clinical use for almost 40 years. Its overall pharmacological profile differs from each of the other classes of psychotropic drugs, of which it bears some resemblance to their action, i.e., benzodiazepines, antipsychotic phenothiazines, and tricyclic antidepressants [25]. In vitro, trazodone is a weak but specific inhibitor of the synaptosomal uptake of serotonin [25], and like mianserin and the classical presynaptic adrenoreceptor yohimbine, it binds to a1- and a2-adrenoreceptor sites [26]. Whereas mianserin shows an equal affinity for a1 and a2 sites, trazodone is about five-fold more active at the a1-sites. In vivo, trazodone reacts with central serotonin receptors in the frontal cortex, an antagonistic activity like that of mianserin [26,27]. Moreover, trazodone has been shown to bind to opioid receptors as well, but only at much higher concentrations than those needed for the interaction with the serotonin- and noradrenergic receptors [28].

Venlafaxine is a structurally phentylethylamine antidepressant drug. In vitro, venlafaxine blocks the synaptosomal uptake of both noradrenaline and serotonin and, to a lesser degree, dopamine [29]. The concentrations of venlafaxine that are necessary to block the reuptake of these monoamines by 50% (IC50) are 0.21 mmol/L for serotonin, 0.64 mmol/L for noradrenaline, and 2.8 mmol/L for dopamine [30,31]. Venlafaxine has not been shown to inhibit MAO [31]), and its monoamine-inhibitory properties differ from those of the serotonin-selective reuptake inhibitors, which show high selectivity for serotonin reuptake only [30]. Moreover, venlafaxine is inactive as a ligand in vitro to a1-, a2-adrenoreceptors, muscarinic, and histaminergic receptors [30,31,32,33,34] and no data has been published regarding its binding to opioid receptors.

Reboxetine inhibits noradrenaline reuptake in vitro to a similar extent as the tricyclic antidepressant desmethylimipramine. It does not affect dopamine or serotonin reuptake, and it has a low (both in vivo and in vitro) affinity for adrenergic, muscarinic, cholinergic, histaminergic, dopaminergic, and serotonergic receptors [35].

Moclobemide is a RIMA compound—a reversible inhibitor of mono-amine oxidase, type A. The nonreversible, ‘classical’ monoamine oxidase-inhibitors (MAO-Is) are a group of efficacious antidepressant medications serendipitously discovered in the early 1950s but hardly prescribed nowadays owing to their severe adverse effects, particularly the hypertensive reactions which may even culminate in a full-blown cerebrovascular accident [36]. It was only with the introduction of the reversible MAOA-I (RIMAs) that interest was reawakened in this group of drugs [37,38]. Moclobemide is an antidepressant that affects the monoaminergic cerebral neurotransmitter system through the reversible inhibition of MAO, preferentially of type A, an action that decreases the metabolism of noradrenaline, dopamine, and serotonin and increases their extracellular concentrations [39]. 

The serotonin-selective reuptake inhibitors (SSRIs) are a structurally heterogeneous group of drugs, the first of which (fluoxetine) was introduced in 1988 (followed by fluvoxamine in the same year) as a new class of antidepressants with a favorable side-effect profile [40]. Over the years, fluoxetine, fluvoxamine, sertraline, and paroxetine have been found to possess secondary binding properties (i.e., dopamine reuptake inhibition, muscarinic cholinergic antagonism, noradrenaline reuptake inhibition, nitric oxide synthase inhibition, etc.); thus they are not so selective as initially thought [41]. However, citalopram and its S(+)-enantiomer (escitalopram) have been reconfirmed as the purest SSRIs [42,43].

In a series of previous studies, we evaluated the antinociceptive effects of these eight antidepressant medications (mianserin, mirtazapine, trazodone, venlafaxine, reboxetine, moclobemide, fluvoxamine, and fluoxetine) and the interaction of each one of them with the opioid system in mice. We found that some of these medications exert their antinociceptive effect involving the opioid receptors, while others have no antinociceptive effect or do have, but without any opioid involvement. 

In this present paper, we focus on the presence or absence of the interaction of eight antidepressants with the opioid system in mice and discuss the possible effects of this interaction (when present) on the efficacy of treatment of severe depression, ‘treatment-resistant depression’ (TRD). 

## 2. Results

### 2.1. Dose-Response Curves

#### 2.1.1. Antidepressants with Opioid Interaction

##### Mianserin Antinociceptive Effect

We evaluated the antinociceptive effect of mianserin on mice using a hotplate assay. At doses from 1 to 25 mg/kg, mianserin administered ip produced an antinociceptive effect in the hotplate test in a dose-dependent manner (Figure 1). The antinociceptive effect observed with 1 mg/kg was 12%, while its effect observed with 25 mg/kg mianserin raised to 72%. As the mianserin dose increased beyond 30 mg/kg, the hotplate latencies declined to baseline, yielding a biphasic dose-response curve. 

##### Mirtazapine Antinociceptive Effect

Screening of mirtazapine in mice demonstrated its efficacy as an antinociceptive agent in the hotplate assay. Therefore, we evaluated the antinociceptive effect of mirtazapine on mice in this analgesic assay. At doses from 1 to 7.5 mg/kg, mirtazapine administered ip produced an antinociceptive effect in the hotplate test in a dose-dependent manner. The antinociceptive effect observed with 1 mg/kg was 20%, while its effect at 7.5 mg/kg rose to 70% as the mirtazapine dose increased. When the mirtazapine dose increased beyond 10 mg/kg, the hotplate latencies declined to baseline, producing a biphasic dose-response curve (Figure 1).

##### Trazodone Antinociceptive Effect 

Trazodone induced a dose-dependent antinociceptive effect following ip injection. The ED_50_ for mice in the hotplate assay for trazodone was 24.8 mg/kg (Figure 1). 

##### Venlafaxine Antinociceptive Effect 

Venlafaxine induced a dose-dependent analgesic effect following ip administration (Figure 1). The ED^50^ for mice in the hotplate assay for venlafaxine was 46.7 mg/kg. 

#### 2.1.2. Antidepressants with No Opioid Interaction

##### Reboxetine Antinociceptive Effect

Reboxetine induced a weak antinociceptive effect (Figure 2). Reboxetine reached its maximal effect of 30% analgesia at 10 mg/kg.

##### Moclobemide Antinociceptive Effect

The analgesic effect of moclobemide is shown in Figure 2. Moclobemide administered ip produced a dose-dependent antinociceptive effect with a median effective dose (ED50) of 69.1 mg/kg.

##### Fluoxetine Antinociceptive Effects

The intraperitoneal injection using fluoxetine was almost inactive (at a dose of 25 mg/kg, only 30% of the mice were analgesic (Figure 2). 

##### Fluvoxamine Antinociceptive Effect

Groups of mice were injected with various doses of fluvoxamine, and a clear antinociceptive effect was evident in the hotplate assay. Following ip injection, fluvoxamine elicited analgesia dose-dependently, with an ED^50^ value of 6.4 mg/kg (Figure 2). 

### 2.2. The Sensitivity of Antidepressant Drugs to the Antinociceptive Effect of Selective Opioid Receptor Antagonists

#### 2.2.1. Antidepressants with Opioid Interaction

##### The Sensitivity of Mianserin Antinociceptive Effect to Selective Opioid Receptor Antagonists 

The antinociceptive effect of mianserin was antagonized by naloxone (1 mg/ kg sc; *p* < 0.05), implying that there is an opioid mechanism of action involved in the mianserin-induced antinociceptive effect (Figure 3). At the next stage, the involvement of the selective antagonists of μ, δ, and κ receptors was assessed to evaluate the potential involvement in the mianserin antinociceptive effect. We examined several selective antagonists (Figure 3). We found that β-FNA reversed the mianserin antinociceptive effect at the same dose that morphine analgesia was reversed, suggesting a role for the μ-receptors in the mianserin antinociceptive effect (*p* < 0.05; Figure 3). The dose of naltrindole that reversed DPDPE analgesia only partially reversed the mianserin antinociceptive effect. NorBNI reversed the mianserin antinociceptive effect at the same dose it reversed κ1 analgesia, mediated by U50,The 8H (*p* < 0.05; Figure 3). When taken together, the sensitivity of the mianserin antinociceptive effect to selective antagonists implies μ and δ mechanisms of action and, to a lesser extent—κ effects. 

##### The Sensitivity of the Mirtazapine Antinociceptive Effect to Selective Opioid Receptor Antagonists

The analysis of naloxone (10 mg/kg, sc) showed a significant drug-by-dose interaction [F(2, 114) = 6.02, *p* < 0.01]; the posthoc analysis revealed that the proportion of antinociceptive subjects differed at 10 mg/kg (*p* < 0.01), implying that there is an opioid mechanism of action involved in the mirtazapine-induced antinociceptive effect. At the next stage, the involvement of the selective antagonists of μ, δ, and κ receptors was assessed to evaluate their potential involvement in the mirtazapine antinociceptive effect. We examined several selective antagonists. The doses of naltrindole and NorBNI that receptively reversed DPDPE and U50,488H antinociception, reversed the antinociceptive effect of mirtazapine (*p* < 0.05, Figure 3). In addition, we found that β-FNA given at a dose that blocked morphine antinociception failed to antagonize the mirtazapine antinociceptive effects. The activity of each of the antagonists was confirmed with its prototypic agonists. None of the antagonists mediated antinociception by themselves, nor did they change the baseline latencies of the pretreated animals. In summary, the sensitivity of the mirtazapine antinociceptive effect to selective opioid antagonists implies δ- and κ-mechanisms of action and, to a lesser extent, μ-mechanisms.

##### The Sensitivity of the Trazodone Antinociceptive Effect to Selective Opioid Receptor Antagonists 

The opioid antagonist naloxone (10 mg/kg, sc) inhibited the antinociceptive effect induced by trazodone from 70% to 10% (*p* < 0.01; Figure 3), implying an opioid mechanism of action involved in the trazodone-induced antinociceptive effect. At the next stage, the involvement of the selective antagonists of the mu, delta, and kappa receptors was assessed to evaluate the potential involvement in the trazodone antinociceptive effects. We examined selective antagonists’ dosage of β-FNA, naltrindole, and NorBNI that respectively reversed morphine, DPDPE, and U50,488H antinociception. β-FNA (at the same dose needed to reverse morphine analgesia) reversed the trazodone antinociceptive effect (*p* < 0.001; Figure 3). Naltrindole (at the same dose of that reversed DPDPE analgesia) only partially reversed the trazodone antinociceptive effect. NorBNI did not reverse the trazodone antinociceptive effect at the same dose; it reversed antinociception mediated by 

U50,488H (Figure 3). The activity of each of the antagonists was confirmed with its prototypic agonists. All these antagonists do not mediate antinociception by themselves and do not change the latencies of the baselines of the pretreated animals. The sensitivity of the trazodone antinociceptive effect selective antagonists implies μ-mechanisms of action and, to a lesser extent, δ-mechanisms with no κ effects.

##### The Sensitivity of Venlafaxine Antinociceptive Effect on Selective Opioid Receptor Antagonists 

The antinociceptive effect of venlafaxine was antagonized by naloxone from 80% to 20% (1 mg/kg sc; *p* < 0.005), implying that there is an opioid mechanism of action involved in the venlafaxine-induced antinociceptive effect (Figure 3). 

Next, we assessed the potential involvement of the selective antagonists for μ, δ, and κ opioid receptor subtypes in venlafaxine antinociception (Figure 3). When administered 24 h before testing, β-FNA (40 mg/kg, sc) was found to be a selective μ antagonist. Similarly, the δ selective antagonist naltrindole (20 mg/kg, sc) blocks δ analgesia, and norBNI (10 mg/kg, sc) is a selective κ antagonist. All these antagonists do not mediate antinociception by themselves and do not change the latencies of the baselines of the pretreated animals. NorBNI reversed the venlafaxine antinociceptive effect at the same dose; it reversed κ analgesia mediated by U50,488H (*p* < 0.005; Figure 3). β-FNA (at the same dose needed to reverse morphine analgesia) and naltrindole (at the same dose of that reversed DPDPE analgesia) only partially reverse the venlafaxine antinociceptive effect. The activity of each of the antagonists was confirmed with its prototypic agonists. The sensitivity of the venlafaxine antinociceptive effect to selective antagonists implies a κ- mechanisms of action and, to a lesser extent, μ and δ mechanisms.

#### 2.2.2. Antidepressants without Opioid Interaction

##### Antinociceptive Effect of Opioid Antagonists on Reboxetine Analgesia

The antinociceptive effect of reboxetine (10 mg/kg) was antagonized by naloxone (10 mg/kg sc; *p* < 0.05), implying that there is a weak opioid mechanism of action involved in the reboxetine-induced antinociceptive effect (Figure 4). In the next stage, the involvement of the selective antagonists of μ, δ, and κ receptors was assessed to evaluate their potential involvement in reboxetine’s antinociceptive effect. β-FNA (μ antagonist), naltrindole (δ antagonist), and Nor-BNI (κ antagonist) completely abolished the weak reboxetine antinociceptive effect, suggesting a nonspecific role for the opioid receptors in the reboxetine antinociceptive effect (*p* < 0.05; Figure 4).

##### Antinociceptive Effect of Opioid Antagonists on Moclobemide Analgesia

Effects of pretreatment with naloxone, a nonselective opioid antagonist, on the antinociceptive activity of moclobemide are shown in Figure 4. The analgesia induced by moclobemide (300 mg/kg) was not significantly inhibited by naloxone. The lack of effect for the other antagonists suggests a minor role (if any) for the opioid in moclobemide analgesia. 

##### Antinociceptive Effect of Opioid Antagonists on Fluoxetine Analgesia

Naloxone, at 10 mg/kg, sc, did not abolish the fluoxetine antinociceptive effect. The fact that naloxone could not antagonize the fluoxetine antinociceptive effect indicates that fluoxetine analgesic effects are mediated by a non-opioid mechanism of action (Figure 4).

##### Antinociceptive Effect of Opioid Antagonists on Fluvoxamine Analgesia

Fluvoxamine (30 mg/kg, ip) produced analgesia in 80% of the mice. When fluvoxamine was co-administered with naloxone (10 mg/ kg, sc), 60% of the mice produced analgesia despite the use of naloxone. The fact that naloxone could not antagonize fluvoxamine analgesia indicates that fluvoxamine antinociceptive effects are mediated by a non-opioid mechanism of action (Figure 4).

### 2.3. The Sensitivity of Antidepressant Drugs to the Antinociceptive Effect of Selective Opioid Receptor Agonists

#### 2.3.1. Antidepressants with Opioid Interaction

##### Mianserin Action on Selected Opioid Receptor Subtypes: Agonists

Mianserin—morphine (μ subtype) interactions: we gave the selective agonists of the μ subtype morphine with or without an inactive dose of mianserin (0.5 mg, ip, Table 1). We found approximately an eight-fold shift to the left in the dose-response curve (*p* < 0.05). The ED50 of morphine without mianserin was 4.5 mg/kg, sc, and with mianserin, it was 0.6 mg/kg, sc. 

Mianserin—DPDPE (δ subtype) interactions: we gave the selective agonists of the δ subtype DPDPE with or without an inactive dose of mianserin (0.5 mg, ip, Table 1). We found an almost three-fold shift to the left in the dose-response curve, which did not reach statistical significance. The ED_50_ of DPDPE without mianserin was 298 ng, it, and with mianserin was 132 ng, it.

Mianserin—U50,488H (κ subtype) interactions: we gave the selective agonists of the κ subtype U50,488H with or without an inactive dose of mianserin (0.5 mg, ip, Table 1). We found an almost 10-fold shift to the left in the dose-response curve (*p* < 0.05). The ED^50^ of U50,488H without mianserin was 4.8 mg/kg, sc, and with mianserin, it was 0.5 mg/kg, sc. These results suggest that mianserin, when administered together with opiates, significantly potentiates antinociception mediated by μ and κ opioid receptor subtypes and, to a lesser extent, by δ mechanisms.

##### Mirtazapine Action on Selected Opioid Receptor Subtypes: Agonists

Mirtazapine–morphine (μ subtype) interactions: we gave the selective agonists of the μ subtype morphine with or without an inactive dose of mirtazapine (0.25 mg, ip, Table 1). We found no significant shift in the dose–response. The ED^50^ of morphine without mirtazapine was 5.3 mg/kg, sc, and with mirtazapine, it was 1.9 mg/kg, sc. 

Mirtazapine–DPDPE (δ subtype) interactions: we gave the selective agonists of the δ subtype DPDPE with or without an inactive dose of mirtazapine (0.25 mg, ip, Table 1). No differences were found between the groups. The ED^50^ of DPDPE without mirtazapine was 307 ng, it, and with mirtazapine, it was 236 ng, it.

Mirtazapine–U50,488H (κ subtype) interactions. we gave the selective agonists of the κ subtype U50,488H, with or without an inactive dose of mirtazapine (0.25 mg, ip, Table 1). No differences were found between the groups. The ED^50^ of U50,488H without mirtazapine was 4.4 mg/kg, sc, and with mirtazapine, it was 3.1 mg/kg, sc. These results suggest that mirtazapine, when administered together with opiates, significantly potentiates antinociception mediated by weak μ opioid receptor subtypes and, to less extent, δ and κ mechanisms.

##### Trazodone Action on Selected Opioid Receptor Subtypes: Agonists 

Multiple doses of this did not reach statistical significance. A selective agonist of the mu-subtype morphine was co-administrated with a vehicle or with an inactive (sub-threshold) dose of trazodone (2.5 mg/kg, ip, Table 1). A significant 14-fold shift to the left in the dose-response curve was observed. The ED^50^ of morphine without trazodone was 5.8 mg/kg, and with trazodone, it was 0.4 mg/kg (*p* < 0.005).

Various doses of the selective agonist of the delta subtype DPDPE were injected with or without an inactive dose of trazodone (2.5 mg/kg, ip, Table 1). A two-fold shift to the left in the dose-response curve was detected, which did not reach significance. The ED^50^ of DPDPE without trazodone was 318 ng, it, and with trazodone, it was 167 ng, it. 

The selective agonist of the kappa subtype U50,488H was injected with or without an inactive dose of trazodone (2.5 mg/kg, ip, Table 1). A nonsignificant three-fold shift to the left in the dose-response curve was obtained. The ED^50^ of U50,488H without trazodone was 4.4 mg/kg, sc, and with trazodone, it was 1.4 mg/kg, sc. These results suggest that trazodone, when administered together with opiates, significantly potentiates antinociception mediated by the μ opioid receptor subtype.

##### Venlafaxine Action on Selected Opioid Receptor Subtypes: Agonists

In the next stage, multiple doses of the selective agonist of the μ subtype morphine were co-administered with a vehicle or with an inactive (subthreshold) dose of venlafaxine (2.5 mg/kg, ip; Table 1). A three-fold shift to the left in the dose-response curve was observed. The differences, however, did not reach statistical significance. The ED_50_ of morphine without venlafaxine was 5.8 mg/kg, and with venlafaxine, it was 1.8 mg/kg. Various doses of the selective agonist of the δ subtype DPDPE were injected with or without an inactive dose of venlafaxine (2.5 mg/kg, ip, Table 1). Almost a four-fold shift to the left in the dose-response curve was detected (*p* < 0.005). The ED_50_ of DPDPE without venlafaxine was 320 ng (it), and with venlafaxine, it was 90 ng, it. The selective agonist of the κ subtype U50,488H was injected with or without an inactive dose of venlafaxine (2.5 mg/kg, ip Table 1). An almost six-fold shift to the left in the dose-response curve (*p* < 0.005) was obtained. The ED_50_ of U50,488H without venlafaxine was 5.7 mg/kg, sc, and with venlafaxine, it was 1.0 mg/kg, sc. 

These results suggest that venlafaxine, when administered together with opiates, significantly potentiates antinociception mediated by δ-opioids and κ receptors subtypes, and to a lesser extent, by μ receptors.

#### 2.3.2. Antidepressants without Opioid Interaction

##### Reboxetine Action on Selected Opioid Receptor Subtypes: Agonists

Groups of mice (n ≥ 15 in each group) were injected with an inactive dose of morphine (a nonselective opioid agonist, 0.5 mg/kg) in addition to increasing doses of reboxetine (0.5–10 mg/kg). No significant increase in analgesia was found when a sub-threshold dose of morphine or clonidine was added to reboxetine. When morphine was added to 0.5 or 4 mg/kg reboxetine, a nonsignificant increase in analgesia was detected, and it disappeared with 10 mg/kg reboxetine. 

##### Moclobemide Action on Selected Opioid Receptor Subtypes: Agonists

When morphine, a selective μ opioid agonist, was administered alone, it produced a dose-dependent antinociceptive effect on the tail-flick test, with an ED_50_ of 5.9 mg/kg, sc. When administered at an inactive dose (0.25 mg/kg) with moclobemide, there was no significant shift in the dose-response curve of moclobemide. The ED_50_ value of moclobemide alone was 69.1 mg/kg, and for moclobemide plus morphine, it was 39.7 mg/kg. 

##### Fluvoxamine Action on Selected Opioid Receptor Subtypes

Fluvoxamine—morphine (μ-subtype) interactions. We gave multiple doses of the selective agonist of the μ-subtype morphine sc with or without an inactive dose of fluvoxamine (0.5 mg/kg, ip, Table 2). We found an almost three-fold shift to the left in the dose—response curve, which did not reach statistical significance. The ED_50_ (of morphine without fluvoxamine was 3.5 mg/kg, sc, and with fluvoxamine, it was 1.4 mg/kg, sc.

Fluvoxamine—DPDPE (δ-subtype) interactions. We gave multiple doses of the selective agonist of the δ-subtype DPDPE, it, with or without an inactive dose of fluvoxamine (0.5 mg/kg, ip, Table 2). The ED_50_ of DPDPE without fluvoxamine was 308 ng, it, and with fluvoxamine, it was 341 ng, it. 

Fluvoxamine—U50,488H (κ-subtype) interactions. We gave multiple doses of the selective agonist of the κ-subtype U50,488H sc with or without an inactive dose of fluvoxamine (0.5 mg/kg, ip, Table 2). We found an almost two-fold shift to the left in the dose—response curve. The ED_50_ of U50,488H without fluvoxamine was 4.9 mg/kg, sc, and with fluvoxamine, it was 1.8 mg/kg, sc. These results suggest that fluvoxamine, when administered together with opiates, does not potentiate antinociception-mediated opioid receptor subtypes.

##### Fluoxetine Action on Selected Opioid Receptor Subtypes 

The next step was designed to study the mechanism by which fluoxetine acts selectively on opioid receptor subtypes. To this aim, several doses of the various subtypes of opioids were injected into mice in the absence or presence of an inactive dose of fluoxetine (0.5 mg/kg, sc). We gave multiple doses of the selective agonist of the μ-subtype morphine with or without an inactive dose of fluoxetine (0.5 mg/kg, sc, Table 2). An almost five-fold shift to the left in the dose–response curve was observed. However, the differences did not reach statistical significance. The ED_50_ of morphine without fluoxetine was 4.9 mg/kg, and with fluoxetine, it was 1.1 mg/kg. 

Various doses of the selective agonist of the δ-subtype DPDPE with or without an inactive dose of fluoxetine (0.5 mg/kg, sc; Table 2) were injected. We found a three-fold shift to the left in the dose-response curve (*p* < 0.005). The ED_50_ of DPDPE without fluoxetine was 318 ng, it, and with fluoxetine, it was 86 ng, it. 

The selective agonist of the κ subtype U50,488H was injected with or without an inactive dose of fluoxetine (0.5 mg/kg, sc; Table 2). We found an almost 10-fold shift to the left in the dose–response curve (*p* < 0.005). The ED_50_ of U50,488H without fluoxetine was 5.1 mg/kg, sc, and with fluoxetine, it was 0.5 mg/kg, sc. 

When administered together with opiates, fluoxetine significantly potentiates analgesia in the δ and κ opioid receptor subtypes and, to a lesser extent, the μ-opioid receptors.

## 3. Discussion

In a series of previous studies, we assessed the antinociceptive effects of several antidepressant medications in a mouse model of acute pain and found that many of the tested drugs exerted antinociceptive effects: some of them involving the opioid system, others had no opioid properties involved in their antinociceptive effects, with others yet, had no antinociceptive effect at all. Beyond the expected activity at the serotonin and noradrenaline receptors (as well as other neurotransmitters recognized as involved in the mechanism of action of some of these medications (i.e., cholinergic receptors, histaminergic receptors, etc.), four drugs exerted clear interaction with opioid receptors (of different types): mianserin, mirtazapine, trazodone, and venlafaxine).

Mianserin alone, when injected ip, elicited an antinociceptive effect in a dose–dependent manner in the hot plate assay at doses from 1–25 mg/kg. When the dose was increased beyond 30 mg/kg, the hotplate latencies returned to baseline, yielding a biphasic dose–response curve, thus suggesting a ‘therapeutic window’ antinociceptive effect. Naloxone abolished this effect, indicating the involvement of opioid mechanisms in mianserin antinociceptive effects. When administered together with various agonists of opioid receptors, mianserin induced a statistically significant potentiation of the m- and the k-opioid receptors and a lower and nonsignificant potentiation of the d-opioid receptor. 

Mirtazapine (injected ip at 1 to 7.5 mg/kg) induced antinociception in a dose-dependent manner, but when doses were increased beyond 7.5 mg/kg, it was found to yield a biphasic dose-response curve, which implies a ‘therapeutic window effect’, much like that of mianserin. Mirtazapine-induced antinociception was antagonized by naloxone, standing for the involvement of the opioid system. When administered together with various opioid receptor agonists, mirtazapine significantly potentiates (mainly) antinociception, mediated by μ- and κ-opioid receptors.

Trazodone induced a dose-dependent antinociceptive effect in the mouse hotplate assay following ip administration. This effect was antagonized by naloxone (implying involvement of the opioid system). When administered together with various opioid receptor agonists, trazodone significantly potentiated antinociception mediated by m- opioid receptors and, only to a lesser extent, d- and k-opioid receptors. 

Venlafaxine induced a clear antinociceptive effect in the mouse hotplate assay. At doses from 1 to 30 mg/kg administered ip, it induced antinociception in a linear dose-dependent manner. This effect was antagonized by naloxone, implying the involvement of the opioid system. When administered together with various opioid receptor agonists, venlafaxine significantly potentiated antinociception mediated by μ-, δ-, and κ-opioid receptor subtypes. 

Reboxetine (ip) induced a very weak, dose-dependent antinociception in the mouse hotplate assay. This weak effect was fully antagonized by all (both selective and nonselective) opioid receptor antagonists, implying a clear (but very weak) nonselective involvement of the opioid system in the antinociceptive properties of reboxetine. When administered together with an inactive dose of an opioid agonist, no significant potentiation in analgesia was noted. 

When the RIMA antidepressant moclobemide was administered ip, it produced a dose-dependent antinociceptive effect (evident at 300 mg/kg). However, it was not significantly inhibited by naloxone, suggesting only a minor (if any) role of the opioid system in moclobemide’s antinociception. 

Fluoxetine (ip) tested in the same hotplate assay yielded only a weak antinociceptive effect, not antagonized by naloxone, implying no involvement of the opioid system. When given with various opioid receptors subtypes agonists, an interaction was found at the k- and d- subtypes only. 

Fluvoxamine (ip) induced a clear antinociceptive effect in the hotplate assay, but this effect was not antagonized by naloxone, implying any involvement of the opioid system. When administered together with various opioid receptors subtypes agonists, no antinociception mediated opioids receptors subtypes were found (Figure 5).

At the end of the 19th century, Emil Kraepelin introduced tincture opium into the treatment of depression, followed by Eugen Bleuler [44]. There is no documentation about the efficacy of the ‘opium cure’ protocol (a three-week procedure during which the daily dose was incrementally increased, then gradually tapered off until discontinued), yet it has been estimated that almost half of the patients could be discharged after this treatment [45,46]. Following the introduction of the MAO-Is and tricyclic antidepressants during the late 1950s, medications without addictive properties, the use of opium for depression was abandoned. A large meta-analysis of about 300 double-blind, randomized controlled studies found that response rates with drugs from different classes ranged from 60 to 68% [47].

Medications with a higher placebo response (i.e., sertraline, 48% placebo response) reached nearly 80% efficacy. All in all, the response rates with the drugs of the newer generations (i.e., SSRIs) were markedly lower than that with the tricyclic antidepressant. [47,48]. Fritze [49] hypothesized that these lower therapeutic response rates might be explained by the newer generation of antidepressants having less affinity with central muscarinic cholinergic receptors. 

Following our speculation that the opioid system is involved in the dysregulation of neurotransmitters, which underly the pathophysiology of severe depression, we successfully augmented the SSRI-failed treatment of refractory depression with naltrexone [50]. We used an opiate antagonist to prevent the hazards of prolonged opiate agonist treatment (e.g., tolerance and physical dependence). Naltrexone is a long-acting, pure competitive opioid antagonist principally of μ-, but also of κ- and δ-opioid receptors in the central nervous system, used mainly in the treatment of both alcohol and opiate dependence [49,51,52] for other conditions such as binge-eating and bulimia nervosa [53], and Prader-Willi syndrome [54]. The hypothesis that interacting with the opioid system (either by an agonist or an antagonist) augments the efficacy of an antidepressant drug derives from studies that found the activity of cerebral mono-aminergic systems (in another animal model) to be regulated to some degree by endogenous opioid input. When that input was chronically blocked, the basal metabolism of monoamines in the brain was not much altered, but the system’s responsiveness to agonist challenge was increased [55]. 

Finally, a recent systematic review and meta-analysis published recently [56] suggests the possibility that buprenorphine, originally introduced as a mixed opioid agonist/antagonist analgesic medication [57], is nowadays prescribed mostly for prolonged, maintenance treatment of people with opiate use disorder (OUD). It may also be a possible treatment for depression [58] as well as suicidal behavior (but may pose certain risks) and concludes that it may have a small benefit for depressive symptoms. The rationale for buprenorphine’s activity as an antidepressant drug (and its ability to counteract the effects of environmental stress) was hypothesized to stem from its actions at the k-opioid receptor [59]. Furthermore, both methadone maintenance treatment (MMT) and buprenorphine/naloxone treatment of opioid use disorder (OUD) have had positive effects on depressive symptoms in patients affected by depression as well [60]. 

Because the use of ketamine and esketamine in the treatment of resistant depression is increasing [61,62] while the hazards of addiction and diversion are no less than those of opioids [63], it is high time to search for new approaches to the connection of the opioid system in the treatment of depression while focusing on the treatment of TRD, i.e., with the biased opioid ligands [64]. 

Controlled clinical studies are needed to establish the efficacy of opiate augmentation (starting first with the antagonist naltrexone) of antidepressants that lack opioid interactions in persons with treatment-resistant depression and the optimal dosage of drugs prescribed.

### 3.1. Limitations

As the results of many basic research studies have failed to be replicated when assessed in higher developmental animals (even more so, later in humans), the translational process from the bench to the bedside (the clinic) is difficult, and, in the case of our studies (with rodents), needs to be first carried out in other large animals (i.e., pigs). Another limitation is the fact that we’ve studied only eight antidepressant medications, while there are many more substances available for clinical use. Finally, we did not assess the possible interaction of the medications with the dopaminergic system, interactions that may contribute to some antinociceptive properties.

### 3.2. Strengths

All eight antidepressant medications were studied using the same strain of mice, and the same approach to the assessment of their antinociceptive effects in the model of nociception was taken. All medication doses were quasi-equipotent of those used in clinical settings, and all opioid agonist and antagonist dosages were used as already described in the literature. This enables us to compare the findings, which is much more complicated when, in most studies, cited regarding tricyclic antidepressants and MAO-I, different rodents and different models were used to characterize the opioid properties of the drugs). 

## 4. Materials and Methods

### 4.1. Subjects and Surgery 

Male ICR mice from Tel-Aviv University colony (Tel-Aviv, Israel), weight 25–35 g, were used. The mice were maintained on a 12 h light: 12 h dark cycle with Purina rodent chow and water available *ad libitum*. Animals were housed in five per cage in a room maintained at 22 °C ± 0.5 °C until testing. Mice were used only once. Central injections were made under light halothane anesthesia using a Hamilton 10 μL syringe fitted to a 30-gauge needle with V1 tubing. Intrathecal (i.t.) injections were introduced by lumbar puncture [65,66]. The Faculty of Medicine Ethical Committee for Animal Experimentation approved all the experiments, which complied with the National Institutes of Health guidelines for animal experimentation of the [DHEW Publication (NIH) 85-23, revised, 1995].

### 4.2. Agents 

Several agents were generously donated as follows: mianserin HCl was a generous gift from Rafa (Jerusalem, Israel); mirtazapine was a generous gift from N.V. Organon (Oss, The Netherlands); trazodone was a generous gift from Unipharm (Ramat-Gan, Israel); venlafaxine HCl was a generous gift from Dexon (Hadera, Israel); moclobemide was generously donated by Hoffman La Roche (Basel, Switzerland); fluoxetine HCl was a generous gift from Eli-Lilly and Company (Indianapolis, IN, USA); fluvoxamine HCL was a generous gift from Agis Laboratories (Yeruham, Israel); morphine by TEVA (Jerusalem, Israel); U50,488H {trans-3,4-dichloro-N-methyl-N-[2-(1-pyrrolindinyl)-cyclohexyl]-benzeneacetamide} by Upjohn Pharmaceutics (West Sussex, UK); naloxone HCl, β-funaltrexamine (β-FNA), (D-Pen2,D-Pen5)enkephalin (DPDPE), naltrindole HCl, nor-binaltorphamine (nor-BNI) were obtained from the research technology branch of NIDA. Reboxetine was purchased from Sigma-Aldrich Israel Ltd. (Rehovot, Israel). Ethrane (Enflurane) was purchased from Abbott (Campoverde, Italy). All other compounds were procured from commercial sources. All the drugs were dissolved in saline. The drugs were prepared immediately before testing, and the requisite doses were given at a volume of 10 mL/kg. All agonists and antagonists were chosen according to our previous studies [67].

None of the antagonists possess any analgesic effect of their own, nor did they alter blood pressure. The agonists studied were used at sub-threshold doses and did not manifest any effect on their own.

### 4.3. Antinociception Assessment

#### 4.3.1. Hot Plate

Mice were tested with the hotplate analgesic meter Model 35D (IITC Inc., Woodland Hills, CA, USA), as previously described [67]. The device basically consists of a metal plate (40 × 35 cm) heated to a constant temperature, on which a plastic cylinder was placed. The analgesic meter was set to a plate temperature of 52.5 ± 0.5 °C. The time of latency was recorded between the second the animal was placed on the hotplate surface until it licked its back paw or jerked it strongly, or jumped out. Baseline latency was determined before experimental treatment for each mouse as the mean of two trials. Post-treatment latencies were determined after 30 min for all the drugs, which were subcutaneously (sc) or intraperitoneally (ip) administered. Post-treatment latencies were determined after 15 min for DPDPE, which was administrated intrathecally (i.t.). To minimize tissue damage, a cut-off time of 30 s was adopted. Antinociception was defined quantitatively as a doubling of baseline values for each mouse. For each dose, at least 10 different animals were checked, and their scores were summarized to determine the percentage of mice with analgesic responses. Each mouse was checked for only one treatment.

#### 4.3.2. Tail-Flick 

Animals were tested for analgesia 30 min after drug injection using the tail-flick method, described in detail elsewhere [46]. Briefly, the baseline latencies were determined before experimental treatment for all animals as the means of two trials. The latency values were between 2.0 and 3.0 s. Post-treatment latencies were determined as indicated for each experiment, and a maximal latency of 10 s was used to minimize tissue damage. Analgesia was defined quantitatively as a doubling or more of baseline values for each mouse [68]. For each dose, at least 10 different animals were checked, and their scores were summarized to determine the percentage of mice with analgesic responses. Each mouse was checked for only one treatment. 

### 4.4. Procedure

As described in the original studies published [66,67,68,69,70,71,72,73,74], the results of the antidepressant drugs were divided into antidepressants with opioid interaction and antidepressants with no opioid interaction. For each drug, the study was conducted over three experiments. 

#### 4.4.1. Experiment 1 

Groups of mice (*n* ≥ 10) were injected intraperitoneally or subcutaneously with different doses, as indicated in the Section 2, to determine the effect of each drug in eliciting antinociception. Normal motor behavior was observed following injection with no sedative effects. The doses of the drugs were chosen based on previous experience regarding the quasi-equipotent of psychotropic drugs in acute pain animal models [42,52]. 

#### 4.4.2. Experiment 2 

The sensitivity of each antidepressant to specific opioid antagonists was examined. First, we determined the effect of the nonselective opioid antagonist naloxone (1 or 10 mg/kg sc). When naloxone inhibited the antinociceptive effect, we further examined the effect of the specific opioid antagonists on each one of the drugs. When analgesia was not antagonized by naloxone (indicating a non-opioid type of antinociception), we did not proceed to test the reaction with specific opioid antagonists. In the specific opioid antagonists experiments, mice administrated with one of the drugs were treated with one of the following drugs: β-FNA (mu antagonist; 40 mg/kg sc) 24 h before the challenge. Naltrindole (delta antagonist) 20 mg/kg sc, nor-BNI (kappa antagonist) 10 mg/kg sc, or saline, were injected simultaneously with the antidepressant drugs. For comparison, β-FNA was tested against morphine, nor-BNI against U50,488H, and naltrindole against DPDPE in separate groups of mice. 

#### 4.4.3. Experiment 3 

The sensitivity of the antidepressant drugs to specific opioid agonists was examined. The action of each drug on selective opioid receptor subtype agonists was tested as follows: (a) groups of mice (*n* ≥ 10) were given increasing doses of morphine, a mu-receptor agonist with an inactive dose of the antidepressant drugs. (b) DPDPE, a selective delta-receptor agonist, was injected intrathecally (i.t.) alone or with an inactive dose of the antidepressant drugs. (c) U50-488H, a selective kappa-receptor agonist, was injected sc alone or with an inactive dose of the antidepressant drugs. 

### 4.5. Statistical Analysis 

Quantitative Dose-response curves were analyzed using an SPSS 18 computer program. This program maximizes the log-likelihood function to fit a parallel set of Gaussian normal sigmoid curves to the dose-response data. The program calculates the dose-response curves and ED_50_ with 95% confidence limits. Single-dose antagonist studies were analyzed using the Fisher exact test. The analysis was performed on the binary antinociceptive response of the mice. In case of significant drug-by-dose interaction, posthoc analysis was performed using the Duncan method.

## Figures and Tables

**Figure 1 ijms-24-11142-f001:**
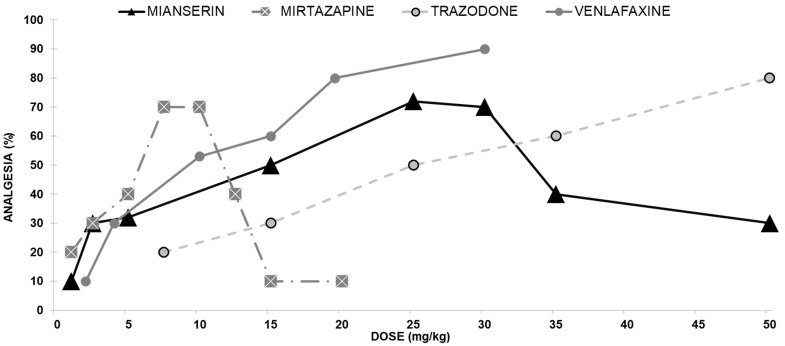
Dose-response curves indicating the antinociceptive effect of each of the four antidepressants exhibiting opioid interaction; Mianserin, Mirtazapine, Trazodone, and Venlafaxine: Each group of mice received an ip injection and were tested with the analgesia hotplate meter test.

**Figure 2 ijms-24-11142-f002:**
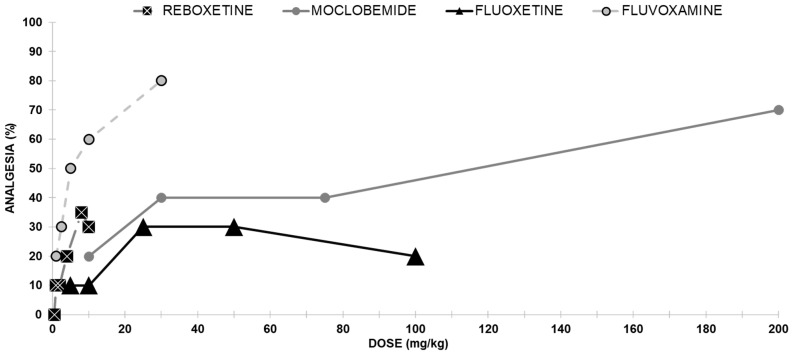
Dose-response curves indicating the antinociceptive effect of each of the four antidepressants without opioid interaction; reboxetine, moclobemide, fluoxetine, and fluvoxamine: each group of mice received an ip injection (except for Reboxetine injected sc) and was tested using the analgesia hotplate meter test (except for moclobemide tested with the mouse-tail flick test).

**Figure 3 ijms-24-11142-f003:**
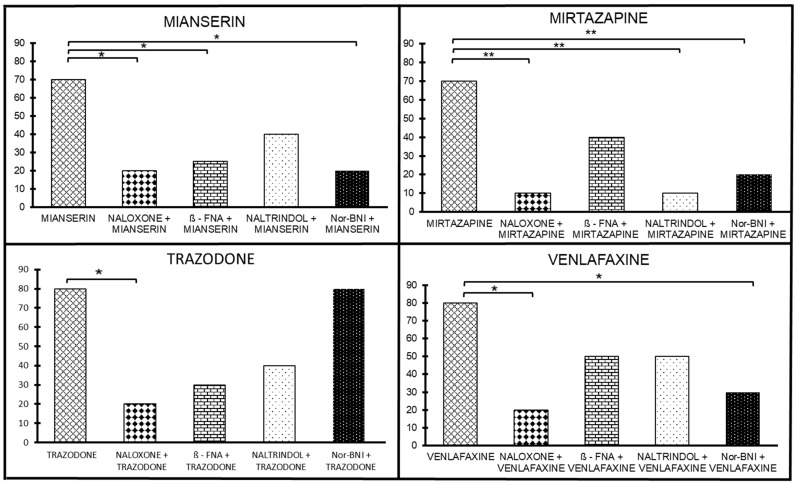
The effect of opioid antagonists on the antinociceptive effect of four antidepressants exhibiting opioid interaction; mianserin, mirtazapine, trazodone, and venlafaxine. Groups of mice were treated with either antidepressant alone or were challenged in addition to naloxone, β-FNA, naltrindole, or nor-BNI. * *p* < 0.05; ** *p* < 0.01.

**Figure 4 ijms-24-11142-f004:**
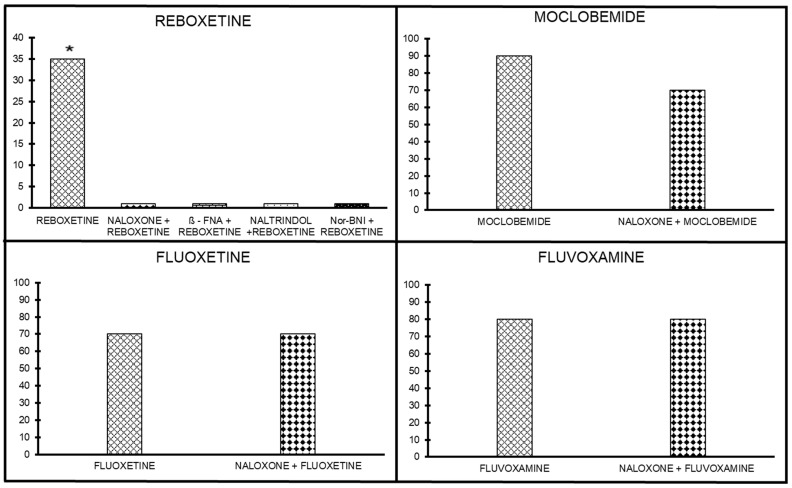
Effects of opioid antagonists on the antinociceptive effect of four antidepressants without opioid interaction; reboxetine, moclobemide, fluoxetine, and fluvoxamine. The effects of naloxone, β-FNA, naltrindole, or nor-BNI can be seen for reboxetine. moclobemide, fluoxetine, and fluvoxamine are seen challenged with naloxone. * *p* < 0.05..

**Figure 5 ijms-24-11142-f005:**
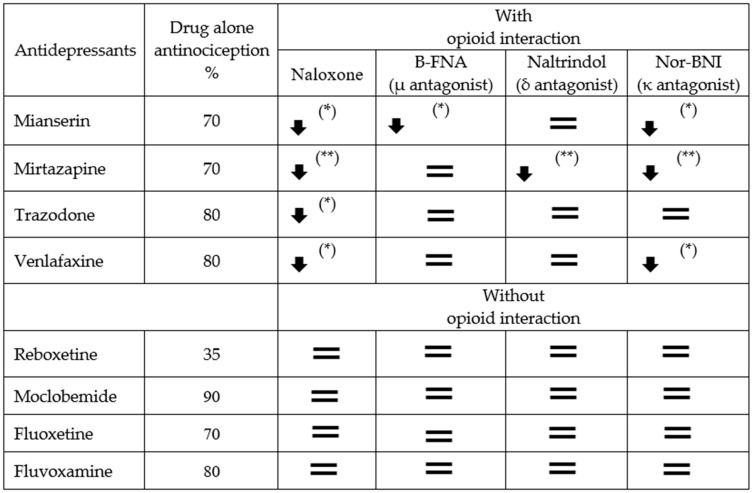
The effect of opioid antagonists on the antinociceptive effect of eight antidepressants: Mianserin, mirtazapine, trazodone, venlafaxine, reboxetine, moclobemide, fluoxetine, and fluvoxamine. The table shows the percentage of antinociception when the drug was administered to the mice alone or when challenged in addition to Naloxone, β-FNA, Naltrindole, or nor-BNI (arrows and equal signs indicate the course of opioid interaction: * *p* < 0.05; ** *p* < 0.01).

**Table 1 ijms-24-11142-t001:** The ED_50_ of the selected opioid receptor subtypes with the absence or the presence of an inactive dose of each of the four antidepressants exhibiting opioid interaction; mianserin, mirtazapine, trazodone, and venlafaxine.

	MIANSERIN			MIRTAZAPINE	
**Opioid receptor subtypes:**	Without MIANSERIN	With MIANSERIN	**Opioid receptor subtypes:**	Without MIRTAZAPINE	With MIRTAZAPINE
Morphine (µ-subtype)	4.5 mg/kg (1.7, 20.7)	0.6 mg/kg (0.04, 0.9)	Morphine (µ-subtype)	5.3 mg/kg (3.5, 10.3)	1.9 mg/kg (1.1, 4.5)
DPDPE (δ-subtype)	298 ng (183, 540)	132 ng (64, 294)	DPDPE (δ-subtype)	307 ng (190.7, 537.8)	236 ng (144.5, 440.8)
U50,488H (K1-subtype)	4.8 mg/kg (2.8, 10.3)	0.5 mg/kg (0.2, 1.2)	U50,488H (K1-subtype)	4.4 mg/kg (2.1, 10.0)	3.1 mg/kg (1.3, 9.8)
	**TRAZODONE**			**VENLAFAXINE**	
**Opioid receptor subtypes:**	Without TRAZODONE	With TRAZODONE	**Opioid receptor subtypes:**	Without VENLAFAXINE	With VENLAFAXINE
Morphine (µ-subtype)	5.8 mg/kg (3.5, 14.6)	0.4 mg/kg (0.22, 0.9)	Morphine (µ-subtype)	5.8 mg/kg (3.5, 14.2)	1.8 mg/kg (0.9, 3.6)
DPDPE (δ-subtype)	318.24 ng (181, 624)	167.5 ng (94.8, 321.9)	DPDPE (δ-subtype)	320 ng (178, 653)	90 ng (49,174)
U50,488H (K1-subtype)	4.4 mg/kg (2.1, 9.8)	1.35 mg/kg (0.6, 3.2)	U50,488H (K1-subtype)	5.7 mg/kg (2.2, 103.2)	1.0 mg/kg (0.4, 0.4)

**Table 2 ijms-24-11142-t002:** The ED_50_ of the selected opioid receptor subtypes with the absence or the presence of an inactive dose of the two antidepressants that did not exhibit opioid interaction: fluvoxamine or fluoxetine.

	FLUOXETINE		FLUVOXAMINE	
Opioid Receptor Subtypes:	Without FLUVOXAMINE	With FLUVOXAMINE	Without FLUVOXAMINE	With FLUVOXAMINE
Morphine (µ-subtype)	4.9 mg/kg (2.8, 13.5)	1.1 mg/kg (0.2, 5.0)	3.5 mg/kg (1.8, 6.8)	1.4 mg/kg (0.8, 2.8)
DPDPE (δ-subtype)	318 ng (200, 568)	86 ng (44, 155)	308 ng (238, 649)	341 ng (200, 567)
U50,488H (K1-subtype)	5.1 mg/kg (2.8, 12.7)	0.5 mg/kg (0.2, 1.2)	4.9 mg/kg (2.8, 10.4)	1.8 mg/kg (0.8, 4.0)

## Data Availability

Not applicable.

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
