# Peer review of "Treatment-Resistant Depression (TRD): Is the Opioid System Involved?"

_ijms, 2023, doi:10.3390/ijms241311142_

Round 1
Reviewer 1 Report
This is a preclinical study which investigated the nociceptive effects of different antidepressants and the relationship between those effects and opioid system. The findings implicate dose-dependent antinociceptive effects of trazodone and venlafaxine, which were antagonized by naloxone. This study contributes to understanding the mechanism of action of antidepressants.
Introduction
TRD patients have significantly more outpatient visits and are at
least twice as likely to be hospitalized-please, provide reference
Please, provide a short description on the subtypes of opioid receptors
However, it seems that mirtazapine’s intrinsic noradrenergic activity counteracts its his
taminergic effects ]20[- this is not clear, please, explain
However, citalopram and its S(+)-enantiomer (escitalopram) have been reconfirmed as the purest SSRIs [39].-citalopram was found to have some antihistaminergic properties, please comment
In a series of previous studies, we evaluated the antinociceptive effects of this eight- antidepressant medication –which eight, please list
Discussion
Four drugs exerted clear interaction with opioid receptors (of different types)- which four drugs?
A large meta-analysis of about 300 double-blind randomized controlled studies found that response rates with drugs from different classes ranged from 60 to 68%. Medications with a higher placebo response (i.e., sertraline, 48% placebo response) reached nearly 80% efficacy (please, add references)
Fritze [56] hypothesized that these lower therapeutic response rates might be explained by the newer generation antidepressants lesser affinity to central muscarinic cholinergic receptors. Please, comment how does anticholinergic activity contribute to antidepressant response?
As the use of ketamine in the treatment of treatment resistant depression is increasing
[69]-please, provide more details. Is ketamine registered as antidepressant in any country? Instead, please mention esketamine, which is registered for TRD
Controlled clinical studies are needed to establish the efficacy of opiate-augmenta
tion of antidepressants that lack opioid interactions-this in unclear. Please, explain which opiates and which antidepressants specifically? How and why can patients benefit from opiate augmentation?
Please, provide limitations and strengths
Author Response
We carefully read the reviewers’ comments and revised the manuscript accordingly.
Following are our replies.

Reviewer 2 Report
The present study focused on the presence or absence of the interaction of eight antidepressants with the opioid system in mice and discuss the possible effects of this interaction (when present) on the efficacy of treatment of severe depression, ‘treatment resistant depression’ (TRD). I think the manuscript new and intriguing contents, however the authors should revise it according to the following concerns;
1. The authors should describe whether or not there are clinical differences of eight antidepressants derived from the differences in relation to the opioid system of each antidepressant.
2. The authors should discuss more in detail on the treatment strategy through the opioid system of TRD based on the present results.
3. The authors should shorten the INTRODUCTION section.
4. The authors should summarize the present results of the interaction of eight antidepressants with the opioid system as a table for readers’ convenience.
fair
Author Response
We have read the reviewers’ comments very carefully and revised the manuscript accordingly.
Following are our replies.

Round 2
Reviewer 2 Report
I think the manuscript has properly been revised according to the reviwer's comments.
fair